# A review of HPV and HBV vaccine hesitancy, intention, and uptake in the era of social media and COVID-19

Emily K Vraga[1]*, Sonya S Brady[2], Chloe Gansen[1], Euna Mehnaz Khan[3], Sarah L Bennis[2], Madalyn Nones[2], Rongwei Tang[1], Jaideep Srivastava[3], Shalini Kulasingam[2]

[1]Hubbard School of Journalism and Mass Communication, University of Minnesota, Minneapolis, United States; [2]Division of Epidemiology and Community Health, University of Minnesota School of Public Health, Minneapolis, United States; [3]Department of Computer Science and Engineering, College of Science and Engineering, University of Minnesota, Minneapolis, United States

**Abstract** Prior to the COVID-19 pandemic, the World Health Organization named vaccine hesitancy as one of the top 10 threats to global health. The impact of hesitancy on the uptake of human papillomavirus (HPV) vaccines was of particular concern, given the markedly lower uptake compared to other adolescent vaccines in some countries, notably the United States. With the recent approval of COVID-19 vaccines, coupled with the widespread use of social media, concerns regarding vaccine hesitancy have grown. However, the association between COVID-related vaccine hesitancy and cancer vaccines such as HPV is unclear. To examine the potential association, we performed two reviews using Ovid Medline and APA PsychInfo. Our aim was to answer two questions: (1) Is COVID-19 vaccine hesitancy, intention, or uptake associated with HPV or hepatitis B (HBV) vaccine hesitancy, intention, or uptake? and (2) Is exposure to COVID-19 vaccine misinformation on social media associated with HPV or HBV vaccine hesitancy, intention, or uptake? Our review identified few published empirical studies that addressed these questions. Our results highlight the urgent need for studies that can shift through the vast quantities of social media data to better understand the link between COVID-19 vaccine misinformation and disinformation and its impact on uptake of cancer vaccines.

*For correspondence: ekvraga@umn.edu

Competing interest: The authors declare that no competing interests exist.

## Editor's evaluation

This manuscript provides a useful literature review on questions regarding the COVID-19 pandemic and the introduction of the COVID-19 vaccine: What is the impact of HPV and HBV vaccination hesitancy as it unfolds in the context of COVID-19 vaccine hesitancy; and how is this relationship reflected in the current era of rampant social media misinformation? The authors' solid review provides insights, partly empirical, regarding the role of HPV and HBV vaccine hesitancy in the modifying context of the pandemic.

## Introduction

The issue of vaccine hesitancy is not a new one. For as long as there have been vaccines, there have been skeptics who distrusted the efficacy and safety of vaccination, and who have shared misinformation about their value. Celebrities such as Jenny McCarthy (*Gottlieb, 2016*; *Largent, 2012*) and news media norms of balance have given such skeptics increased attention (*Clarke, 2008*; *Dixon*

and Clarke, 2013). Such misinformation can be a major driver of vaccine hesitancy (*Enders et al., 2022*; *Lee et al., 2022*). We define misinformation as any information that counters the current best evidence and expert consensus on the topic (*Vraga and Bode, 2020*; see also *Nyhan and Reifler, 2010*; *Southwell et al., 2022*). Given the strong scientific and medical consensus on vaccines, there is often a clear distinction between what is true versus false (i.e., misinformation).

What makes recent years unique in terms of vaccine hesitancy and the spread of misinformation is the emergence of social media. Social media allows for a democratization of voices, with user-generated content appearing alongside (and often with equal prominence to) official scientific and medical voices. Social media, moreover, can be both insular and porous, allowing diverse views to compete, but also for individuals to often identify and interact primarily with those who share their views (*Jones-Jang and Chung, 2022*; *Schmidt et al., 2018*), a phenomenon that has been called the 'echo chamber.' Thus, misinformation can often live much longer in sheltered corners of social media than it might if subjected to greater public scrutiny. Existing studies have demonstrated that many social media platforms are rife with vaccine misinformation (*Suarez-Lledo and Alvarez-Galvez, 2021*), which is concerning because misinformation often outperforms accurate information in terms of popularity and reach (*Vosoughi et al., 2018*). For these reasons, the WHO included vaccine hesitancy as one of the top 10 health challenges facing the globe in 2019, even before the emergence of COVID-19 upended public health (*WHO, 2019*).

COVID-19 created a perfect storm in terms of preexisting vaccine hesitancy and a media environment that was well suited to amplify concerns and misinformation about the development of a COVID-19 vaccine that did not have decades of clear evidence of its specific safety and efficacy. For those who have been engaged in understanding and reducing vaccine hesitancy toward other vaccines, this raises two questions. First, is hesitancy towards the relatively new COVID vaccine – which was developed based on decades of evidence for other vaccines – associated with hesitancy toward other, more established vaccines? Second, is exposure to COVID-19 vaccine misinformation on social media associated with hesitancy toward other vaccines?

We answer these questions in the context of the human papillomavirus (HPV) and hepatitis B (HBV) vaccines because the former has been received with hesitancy by some segments of the public (*Saulsberry et al., 2019*), as was COVID-19. This is due in part to the politicization of these vaccines. We also include HBV vaccination to determine whether the impact extends to other cancer-related vaccines.

## Results

For our first research question, namely whether COVID-19 vaccine hesitancy, intention, or uptake (COVID constructs) are empirically associated with HPV or HBV vaccine hesitancy, intention, or uptake (HPV or HBV constructs), only 7 of the 322 reports reviewed met our criteria. All studies were cross-sectional in nature and examined the associations between hesitancy, intention, or uptake for COVID-19 and HPV vaccines, but not the HBV vaccine. Five of the seven studies found significant associations after adjustment for covariates (i.e., potential confounders), and one found an association in unadjusted analyses.

Three of seven cross-sectional studies framed their questions in terms of whether COVID-19 constructs impacted HPV constructs. In January 2021, *Shimizu et al., 2022* conducted a survey of 1257 Japanese caregivers with daughters aged 12–16 y, recruited via a registered research panel. Among other potential determinants of intention to obtain the HPV vaccine for one's child, the authors measured intention to obtain a COVID-19 vaccine for their child and oneself once the vaccine became available if side effects were 'common' (i.e., not serious). Models were adjusted for demographics, health literacy, media contact, and perceptions and beliefs. Odds of HPV vaccine intention were higher among caregivers who intended to vaccinate their child (odds ratio [OR] = 4.16, 95% CI: 2.79–6.19) and themselves (OR = 1.96, 95% CI: 1.29–3.00) against COVID-19. In March 2021, *Tsui et al., 2023* conducted a survey of 357 parents of adolescents aged 9–17 y who were participating in an academic enrichment program for low-income, first-generation, racial or ethnic minority families in Los Angeles, CA, USA. Among other potential determinants of HPV vaccine hesitancy, the authors measured intention to obtain a COVID-19 vaccine for their children once it became available. In unadjusted models, the odds of HPV vaccine hesitancy were higher among parents who reported being only somewhat likely (OR = 1.58, 95% CI: 0.75–3.31), not too likely (OR = 2.86, 95% CI: 1.16–7.05), or not likely at all (OR = 15.7, 95% CI: 4.45–55.70) to get their children vaccinated against COVID-19,

relative to parents who were very likely to do so. Only the latter association remained significant in models adjusting for covariates, including medical mistrust and exposure to negative information about the HPV vaccine. COVID-19 vaccine intent for children was not associated with HPV vaccine initiation for the youngest child in the family. Using electronic health record data as of November 2021, Coronado and colleagues (2023) examined the association between initiation of the COVID-19 and HPV vaccines among over 40,000 Kaiser Permanente Northwest members aged 12–17 y who lived in Oregon or southwest Washington, USA. Adjusting for age, sex, race, ethnicity, insurance status, urban/rural status, and having a clinic visit in the past year, having initiated the COVID-19 vaccine was associated with greater odds of having initiated the HPV vaccine (OR = 4.01, 95% CI: 3.80–4.23). Not surprisingly, given the cross-sectional study design, having initiated the HPV vaccine also was associated with greater odds of having initiated the COVID-19 vaccine (OR = 4.02, 95% CI: 3.81–4.24).

Four of the cross-sectional studies exclusively framed their questions in terms of whether HPV constructs impacted COVID-19 constructs but were again cross-sectional in nature. Between November and December 2020, *Berenson et al., 2021* surveyed 342 women aged 18–45 y who were recruited from reproductive clinics in South Texas, USA. Adjusting for other factors, women had greater intention to receive a doctor-recommended COVID-19 vaccine if they had previously received the HPV vaccine (OR = 2.26, 95% CI: 1.07–4.79). Between March and April 2021, *Phan et al., 2022* conducted a survey of 513 caregivers recruited from a pediatric healthcare system in the mid-Atlantic United States. Caregivers were diverse with respect to race, ethnicity, socioeconomic status, and rurality/urbanity. History of child's receipt of at least one dose of the HPV vaccine if aged 13–20 y was not associated with caregiver intention to vaccinate their children against COVID-19. Between February and March 2021, Kecojevic and colleagues (2021) surveyed 457 students attending a public university in New Jersey, USA. In unadjusted analyses, being vaccinated against 'other infectious diseases,' with HPV given as an example, was not associated with being vaccinated against COVID-19; however, this variable was associated with greater intention to become vaccinated against COVID-19 (OR = 2.84, 95% CI: 1.84– 4.37). This association became nonsignificant when adjusting for covariates. Between November and December 2021, Ogaz and colleagues (2023) recruited 1039 UK men who have sex with men aged 16 and older from social networking and dating applications to complete surveys. Known HPV vaccination was associated with completed COVID-19 vaccination (OR = 3.32, 95% CI: 1.43–7.75), adjusting for age, ethnicity, gender, sexual orientation, education, employment, relationship status, risk for severe COVID-19 illness based on a medical condition, COVID-19 infection, and self-worth.

Two additional studies did not fully meet criteria for our first research question because they lacked a statistical comparison of the relationship between COVID-19 and HPV (or HBV) constructs, but did not explore COVID-19 and HPV constructs in relation to one another. In each study, parents were recruited via an online research panel. Between September and October 2020, *Olagoke et al., 2022* conducted a survey of 342 parents of adolescents aged 11–17 y who had never been vaccinated against HPV. To meet eligibility criteria, parents had to identify as Christian and live in the United States. In regression analyses adjusting for sociodemographic variables, perceived vulnerability of one's child to HPV ($\beta$ = 0.32, 95% CI: 0.21–0.44) and perceived response efficacy of the HPV vaccine ($\beta$ = 0.41, 95% CI: 0.28–0.53) were independently associated with greater intent to vaccinate one's child against COVID-19, while perceived severity of HPV was not associated with this outcome ($\beta$ = 0.16, 95% CI: –0.01 to 0.32). All three HPV constructs – perceived vulnerability of one's child to HPV ($\beta$ = 0.37, 95% CI: 0.25–0.48), perceived response efficacy of the HPV vaccine ($\beta$ = 0.46, 95% CI: 0.33– 0.59), and perceived severity of HPV ($\beta$ = 0.21, 95% CI: 0.05–0.38) – were independently associated with parents' intention to vaccinate oneself against COVID-19. Although intention to vaccinate one's child against HPV was measured, it was not examined as a predictor of COVID-19 vaccination intentions. In the second study, performed in August 2021, *Manganello et al., 2023* conducted a survey of 452 parents of children aged 9–14 y living in different communities across the United States. Among parents who would be likely to vaccinate their child against COVID-19, 75% would also be likely to vaccinate their child against HPV. Conversely, among parents who would be likely to vaccinate their child against HPV, 58% would also be likely to vaccinate their child against COVID-19. Although no statistical test was conducted, results suggested that there was greater hesitancy for the COVID-19 vaccine than the HPV vaccine among parents who were accepting of the contrasting vaccine.

Turning to our second research question, our review found that none of the 72 studies reviewed met all three criteria for our second research question, that is, whether exposure to COVID-19 vaccine misinformation on social media was associated with HPV or HBV vaccine hesitancy, intention, or uptake. Studies identified as potential candidates by our literature search generally described COVID-19 vaccine misinformation on social media, without examining whether exposure to misinformation on social media was associated with HPV or HBV vaccine hesitancy, intention, or uptake among individuals within social media networks.

## Discussion

Our research suggests that there is a dearth of published peer-reviewed research studies addressing the question of whether COVID-19 vaccine hesitancy or misinformation spills over to hesitancy toward HPV/HBV vaccines. In our review of studies containing keywords related to both COVID-19 and HPV or HBV vaccine hesitancy, intention, or uptake (RQ1), only seven studies examined the empirical association between COVID-19 and HPV constructs. Moreover, these studies were all cross-sectional in nature, so that any association between vaccine hesitancy, intention, or uptake for both vaccines that was uncovered does not inform the directionality of this effect. However, knowing there is a relationship between hesitancy, intention, or uptake for the COVID-19 and HPV vaccines is instructive. This suggests that there is a potential spillover that should be examined through longitudinal research. Since COVID-19 vaccine has been politicized in the United States (*Motta, 2021*), it is important to further our understanding of how COVID-19 vaccine hesitancy may not only be associated with but also *impact* attitudes toward other vaccines, including those that protect against cancer. Of note, no studies tested the association between COVID-19 constructs and HBV constructs.

Additionally, there were no peer-reviewed studies that explored the empirical association between exposure to COVID-19 *misinformation* on social media and HPV and HBV hesitancy, intention, or uptake (RQ2). This lack of research is very concerning, especially since so much of the populace relies on social media as a primary information source (*Pew, 2022*), and social media use is associated with vaccine hesitancy for multiple vaccines (*Dunn et al., 2017*; *Jennings et al., 2021*). This area is prime for new empirical research, and such findings can inform policymaking and regulations to ensure that social media platform policies address health information to promote public health. In addition, previous research in other domains (e.g., climate change) has found that there can be a reinforcing spiral between science skepticism and information consumption behaviors over time (*Feldman et al., 2014*). Thus, those who are skeptical of vaccines may seek confirmation of their (inaccurate) beliefs and become even more skeptical over time, as predicted by the ideological health spirals model (*Young and Bleakley, 2020*).

### Future research directions based on the systematic review

Based on the literature reviewed, there are a number of questions that need to be answered regarding the potential for spillover effects across vaccines in terms of hesitancy, intention, or uptake. Given scholarly and public concerns that COVID vaccine attitudes are impacting uptake for other vaccines (*Larson et al., 2022*; *Messerly and Mahr, 2022*) and alarms about the rising polarization in the United States surrounding other vaccines (*Frankovic, 2021*), solid empirical research is necessary to validate a potential link between hesitancy, intention, and especially uptake across vaccines.

Our systematic review also points to the need for innovative work to explore the impact of social media exposure to COVID-19 misinformation on HPV and HBV vaccine attitudes. Existing research is often limited to documenting the prevalence of vaccine misinformation on social media (e.g., *Suarez-Lledo and Alvarez-Galvez, 2021*; *Wang et al., 2019*) or looking at the association between social media use and vaccine attitudes (*Dunn et al., 2017*; *Jennings et al., 2021*). Our review found no studies that attempted to empirically relate COVID-19 vaccine misinformation with HPV or HBV vaccine hesitancy, intention, or uptake. Explicitly linking social media misinformation exposure to individual vaccine beliefs and behaviors will require sophisticated efforts to link online exposure to offline health outcomes.

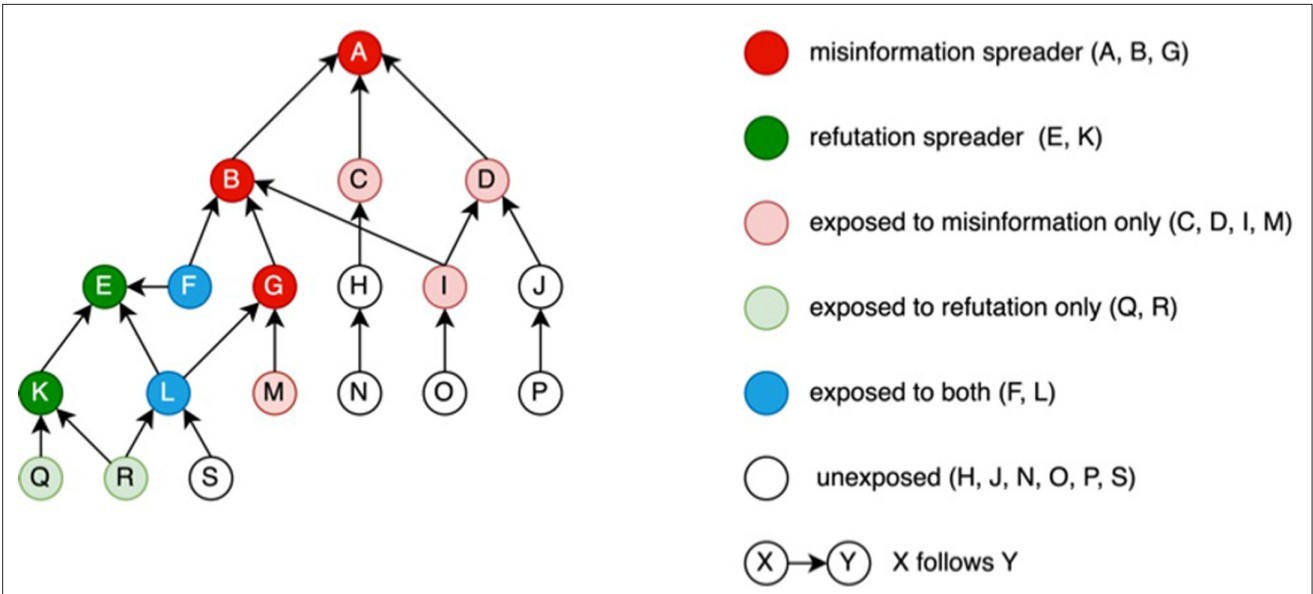

**Figure 1.** Misinformation and refutation propagation network.

## Additional research directions

Innovative, ongoing work in computational methods, epidemiology, and communication research can be leveraged to understand vaccine spillover effects (e.g., COVID-19 vaccine hesitancy leading to HPV hesitancy) and how social media misinformation may contribute to such spillover. For example, we need more sophisticated methods to map how misinformation, refutation of misinformation, and accurate information about different vaccines propagates through a social network. One method is to form a tree structure with an initial source of information depicted as the root (*Figure 1*; *Ma et al., 2017*; *Wu et al., 2015*). Individuals who share misinformation, refutation of misinformation, and accurate information are depicted through different colors throughout the tree structure. Different actors within the social networks may choose to further spread misinformation, refute misinformation, or deliver accurate information. Those who refute misinformation are of particular interest. These are the individuals who may be able to interrupt the spread of misinformation within a network.

To our knowledge, however, investigators have not examined the spread of both vaccine misinformation and refutation in a social network, and whether exposure to both types of information influences vaccine hesitancy, intentions, or uptake. Given the pervasiveness of social media, and the reliance on one's social media network members to inform decision-making, this type of research is urgently needed. In addition, based on our review, investigators have not attempted to link exposure to COVID-19 vaccine misinformation in a social media network to individual-level HPV or HBV vaccine hesitancy, intention, or uptake.

Other future research directions could involve exploring differences in the spread of misinformation on different social media platforms, testing individual and community differences in vulnerability to misinformation spread through social media, and modeling the impact of exposure to misinformation for one vaccine on hesitancy for other vaccines. For example, does the social media platform (e.g., Twitter, Facebook, Instagram, TikTok) impact vaccine hesitancy differently? Does COVID-19 misinformation impact its recipients in different ways depending on the source from which they received it? Are some people more susceptible to believing vaccine misinformation and how can we assess their vulnerability? What is the direction of spillover with respect to hesitancy for two vaccines (e.g., COVID-19 vaccine hesitancy impacting HPV vaccine hesitancy or vice versa)? Putting these pieces together involves developing and testing a comprehensive model that includes social media exposure, vaccine hesitancy, and vaccine uptake for multiple vaccines. Ideally, data would be collected over an extended period of time.

In conclusion, our systematic review underscores the need for longitudinal research examining potential spillover effects between hesitancy for different vaccines. This is especially urgent with new

**Table 1.** Search strategy and number of identified records.

| Step | Search terms | Medline | PsychInfo |
|---|---|---|---|
| 1 | (Misinformation or disinformation or conspiracy theory or rumor or fake news) | 7168 | 14,611 |
| 2 | (Social media) or (social network and online) or (social network and digital) or (social network and internet) | 35,371 | 72,558 |
| 3 | (COVID) or (SARS-CoV-2) | 359,988 | 48,996 |
| 4 | Vaccine and (hesitancy or uptake or intention) | 16,311 | 5661 |
| 5 | HPV or HBV | 101,348 | 6873 |
| 6 | Research question 1: searches 3, 4, and 5 combined with the Boolean term "and" | 84 | 314 |
| 7 | Research question 2: searches 1, 2, 3, 4, and 5 combined with the Boolean term "and" | 7 | 88 |

vaccines in development to prevent or treat cancer, where uptake can save lives (*Winstead, 2022*). Additionally, our review did not identify any studies that empirically examined whether COVID-19 vaccine misinformation on social media relates to HPV or HBV vaccine hesitancy, intention, or uptake. Tracking people's interaction with misinformation on social media, and its refutation, holds promise for providing novel insights into which people are active participants in misinformation spread, and which are active in stopping its spread (*Khan et al., 2023*). This understanding can help in developing strategies to encourage vaccine uptake by suppressing the efforts of those who spread misinformation and accelerating the efforts of those who spread the truth. Such prevention strategies are only likely to be effective when there is a foundation of rigorous research to guide efforts.

## Methods

A systematic literature review was conducted via the Ovid Medline and APA PsychInfo databases. These public databases cover published studies associated with biomedicine and health (Ovid Medline) and psychological, social, and behavioral sciences (PsychInfo). We developed a multistep search strategy (see *Table 1*) to identify the articles that potentially met the criteria to answer two research questions: (1) Is COVID-19 vaccine hesitancy, intention, or uptake associated with HPV or HBV vaccine hesitancy, intention, or uptake? (2) Is exposure to COVID-19 vaccine misinformation on social media associated with HPV or HBV vaccine hesitancy, intention, or uptake? Search steps #1–5 each consisted of keywords for one dimension of the research questions (e.g., mis-/disinformation; social media). The bottom two rows of *Table 1* show how steps #1–5 were used in combination to address the research questions. Two members of the research team independently completed the multistep search in each database and achieved consistent results on October 22, 2022. A bridge search was performed in June 2023 to incorporate all published articles through May 31, 2023. Search steps for the first research question yielded 398 total records (PsychInfo, n = 314; Medline, n = 84); search steps for the second research question yielded 95 total records (PsychInfo, n = 88, Medline: n = 7) (see *Table 1*).

*Figure 2* summarizes the process of identifying relevant peer-reviewed articles using terminology from the 2020 PRISMA guidelines. *Record* refers to the title and/or abstract of an article indexed in a database, whereas *report* refers to the electronic document containing detailed information about a study (e.g., a journal article, dissertation, etc.) (*Page et al., 2021*).

For the first research question, 380 reports were eligible for assessment after removing duplicate (n = 8) or incomplete (n = 9) records and one report that was unable to be retrieved. To meet the criteria for inclusion, the full paper was reviewed to ensure that the study was an empirical study published in an academic journal and (1) measured COVID-19 vaccine hesitancy, intention, or uptake; (2) measured HPV or HBV vaccine hesitancy, intention, or uptake; and (3) tested the statistical association between both measures. For the purposes of this review, we defined an empirical study as one that had measures of observable data. Based on these criteria, a further 56 reports were excluded that were

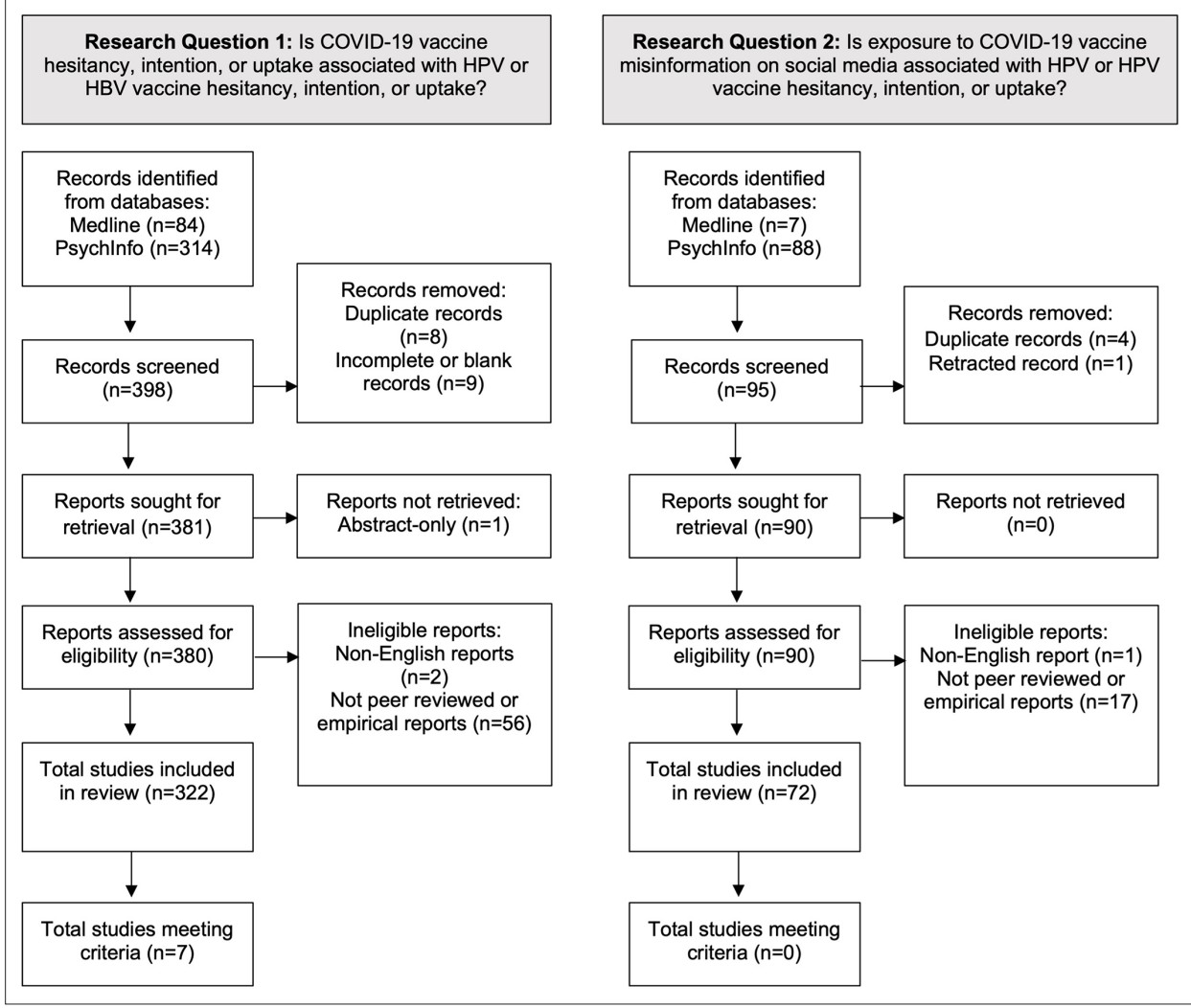

**Figure 2.** PRISMA diagram for full review: identification of peer-reviewed articles. Note: includes articles made available in the databases through May 31, 2023. *Record* refers to the title or abstract of a report indexed in the Medline or PsychInfo database. *Report* refers to an electronic document providing information about a study, such as a journal article or conference abstract (**Page et al., 2021**).

The online version of this article includes the following figure supplement(s) for figure 2:

**Figure supplement 1.** PRISMA figure of the original search.

**Figure supplement 2.** PRISMA figure of the bridge review.

commentaries, reviews, opinion pieces, book chapters, simulation modeling, meta-analyses, or animal studies. Two additional studies were removed that were non-English-language reports, leaving a total of 322 articles classified as eligible for full review. The reports were divided among seven members of the research team to code independently after first meeting to discuss assessment criteria and reaching consensus on the first 16 reports.

For the second research question, 90 reports were eligible for assessment after removing duplicate records (n = 4) and a retracted article (n = 1). To meet the criteria for inclusion, a study had to be an English-language empirical study published in an academic journal and (1) measure exposure to COVID-19 vaccine misinformation (or disinformation, conspiracy theories, rumors, or fake news) on social media; (2) measure HPV or HBV vaccine hesitancy, intention, or uptake; and (3) test the statistical association between both measures. Based on these criteria, a further 18 reports were excluded, leaving a total of 72 articles classified as eligible for full review. Two members of the research team coded all remaining 72 articles after five members of the research team first met to discuss assessment criteria and reach consensus on reports.

## Additional information

### Funding
No external funding was received for this work.

### Author contributions
Emily K Vraga, Shalini Kulasingam, Conceptualization, Formal analysis, Supervision, Investigation, Methodology, Writing – original draft, Project administration, Writing - review and editing; Sonya S Brady, Conceptualization, Data curation, Supervision, Investigation, Methodology, Writing – original draft, Project administration, Writing - review and editing; Chloe Gansen, Formal analysis, Investigation, Visualization, Writing – original draft, Writing - review and editing; Euna Mehnaz Khan, Formal analysis, Investigation, Visualization, Writing – original draft; Sarah L Bennis, Madalyn Nones, Rongwei Tang, Formal analysis, Investigation; Jaideep Srivastava, Conceptualization, Supervision, Methodology, Writing - review and editing

### Author ORCIDs
Emily K Vraga (ID) http://orcid.org/0000-0002-3016-3869
Euna Mehnaz Khan (ID) https://orcid.org/0000-0002-0030-7841
Rongwei Tang (ID) http://orcid.org/0000-0003-3441-276X

### Decision letter and Author response
Decision letter https://doi.org/10.7554/eLife.85743.sa1
Author response https://doi.org/10.7554/eLife.85743.sa2

## Additional files

### Supplementary files
• Supplementary file 1. This supplementary files contains the search results for the original search in October 2022 and the bridge search in June 2023. (**a**) Original search. Conducted on October 22, 2022. Includes articles made available in the databases through October 22, 2022. (b) Bridge review. Conducted on June 6, 2023. Includes articles made available in the databases from October 22, 2022, to May 31, 2023.

• MDAR checklist

### Data availability
Data regarding the articles reviewed is available at: https://doi.org/10.5061/dryad.fttdz08zw.

The following dataset was generated:

| Author(s) | Year | Dataset title | Dataset URL | Database and Identifier |
| --- | --- | --- | --- | --- |
| Vraga EK, Brady SS, Gansen C, Khan E, Bennis SL, Nones M, Tang R, Srivastava J, Kulasingam S | 2023 | A review of HPV and HBV vaccine hesitancy, intention and uptake in the era of social media and COVID-19 | https://dx.doi.org/10.5061/dryad.fttdz08zw | Dryad Digital Repository, 10.5061/dryad.fttdz08zw |

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
