## [Editor Report]

This manuscript provides a useful literature review on questions regarding the COVID-19 pandemic and the introduction of the COVID-19 vaccine: What is the impact of HPV and HBV vaccination hesitancy as it unfolds in the context of COVID-19 vaccine hesitancy; and how is this relationship reflected in the current era of rampant social media misinformation? The authors' solid review provides insights, partly empirical, regarding the role of HPV and HBV vaccine hesitancy in the modifying context of the pandemic.

---

## [Decision Letter]

**Decision letter after peer review:**

Thank you for submitting your article "HPV and HBV vaccine hesitancy, intention and uptake in the era of social media and COVID-19: A review" for consideration by *eLife*. Your article has been reviewed by 2 peer reviewers, and I oversaw the evaluation in my dual role of Reviewing Editor and Senior Editor. The reviewers have opted to remain anonymous.

Essential revisions:

As is customary in *eLife*, the reviewers have discussed their critiques with one another and with the Editors. The decision was reached by consensus. What follows below is my edited compilation of the essential and ancillary points provided by reviewers in their critiques and in their interaction post-review. Please submit a revised version that addresses these concerns directly. Although we expect that you will address these comments in your response letter, we also need to see the corresponding revision clearly marked in the text of the manuscript. Some of the reviewers' comments may seem to be simple queries or challenges that do not prompt revisions to the text. Please keep in mind, however, that readers may have the same perspective as the reviewers. Therefore, it is essential that you amend or expand the text to clarify the narrative accordingly.

*Reviewer #1:*

This study reviews the literature on cancer vaccine hesitancy in relation to more recent COVID-19 vaccine hesitancy. The review is sound and the conclusions are warranted.

Recommendations for the authors:

Be sure to highlight the fact that many studies are not causal, meaning that it remains very unclear if exposure to misinformation causes anti-vaxx or misinformed beliefs, and it remains unclear if false beliefs lead to vaccine refusal. People could be seeking out and believing ideas that comport with actions they would have taken anyway. There has been a well-organized anti-vaxx movement for more than twenty years and many people were vaccine-hesitant long before COVID and HPV vaccines.

*Reviewer #2:*

This paper presents a review to address questions of COVID-19 vaccine hesitancy and its relationship to HPV and HBV vaccine hesitancy, in addition to the influence of social media misinformation on vaccine intentions and uptake. While these are important issues to be addressed, there are several methodological constraints that limit the overall impact of the findings and conclusions. The review search strategy is limited to public databases, Psychinfo and Medline. Other databases could have been addressed as well. A better-defined systematic review would have more clearly defined the search strategy. There is no mention of following PRISMA guidelines. The search strategy as presented lacks details, such as the procedure or individual records. Were titles and abstracts removed as a first step prior to final record selection? These gaps may ultimately have led to the few articles meeting the inclusion criteria for the first question and no articles addressing the second question. In the Results section, a brief description of several studies' limitations was provided, but in the methods, it was indicated that cross-sectional correlational studies were to be targeted, but the lack of longitudinal studies was highlighted as a significant gap. The Discussion section is short on implications and more of a 'Future Directions'. Finally, there is an extensive Future Research Directions on the role of social media that is based on no findings from the systematic and more of a polemic on its role in facilitating the dissemination of misinformation on vaccines. While this is interesting, the model presented in Figure 2 on the links among misinformation propagators and refuters does not really add to the overall impact of the paper, which should be based on the findings.

Recommendations for the authors:

1. Introduction could be more concise and pointed, and many of the statements made are not directly relevant to the research questions.

2. Expand the search strategy using consultation from a librarian and under PRIMA guidelines and registration. This will strengthen the fundamental search strategies and might aid in modifying the two objectives of the study.

3. The Methods should be more clearly defined in a stepwise fashion, e.g., including first a title and abstract screening to identify articles for full-text review.

4. Be clear in the limits of your investigation- only cross-sectional studies and/or longitudinal studies and support these inclusion criteria.

5. In the discussion, please ensure that the implications of summarized studies are made clear, especially regarding any studies ultimately discovered regarding the role of social media information if found in a more expanded systematic review.

6. Finally, ensure that the future directions are based soundly on the findings and any additional innovative ideas that spring from the author's own insights or experience.

---

## [Author Response]

Essential revisions:As is customary in eLife, the reviewers have discussed their critiques with one another and with the Editors. The decision was reached by consensus. What follows below is my edited compilation of the essential and ancillary points provided by reviewers in their critiques and in their interaction post-review. Please submit a revised version that addresses these concerns directly. Although we expect that you will address these comments in your response letter, we also need to see the corresponding revision clearly marked in the text of the manuscript. Some of the reviewers' comments may seem to be simple queries or challenges that do not prompt revisions to the text. Please keep in mind, however, that readers may have the same perspective as the reviewers. Therefore, it is essential that you amend or expand the text to clarify the narrative accordingly.Reviewer #1:This study reviews the literature on cancer vaccine hesitancy in relation to more recent COVID-19 vaccine hesitancy. The review is sound and the conclusions are warranted.Recommendations for the authors:Be sure to highlight the fact that many studies are not causal, meaning that it remains very unclear if exposure to misinformation causes anti-vaxx or misinformed beliefs, and it remains unclear if false beliefs lead to vaccine refusal. People could be seeking out and believing ideas that comport with actions they would have taken anyway. There has been a well-organized anti-vaxx movement for more than twenty years and many people were vaccine-hesitant long before COVID and HPV vaccines.

We thank the reviewer for these suggestions. We have greatly revised the discussion to clearly state that the cross-sectional nature of our studies does not allow us to discern the directionality of any association between COVID-19 vaccine hesitancy, intention, or uptake, COVID-19 misinformation exposure on social media, and HPV/HBV vaccine hesitancy, intention, or uptake. We acknowledge that people skeptical of vaccines may be seeking out confirmation of their beliefs but also point to existing literature suggesting that this can create a self-reinforcing cycle and that is theorized to apply to COVID-19 behaviors. We believe this represents a more clear theoretical model for any association between vaccine beliefs and online behaviors, as well as reinforces the practical importance of empirically testing this relationship.

In addition, we acknowledge in the introduction of the manuscript that vaccine hesitancy is not new or novel, although we point to the ways in which we think social media may be exacerbating vaccine hesitancy and spillover.

Reviewer #2:This paper presents a review to address questions of COVID-19 vaccine hesitancy and its relationship to HPV and HBV vaccine hesitancy, in addition to the influence of social media misinformation on vaccine intentions and uptake. While these are important issues to be addressed, there are several methodological constraints that limit the overall impact of the findings and conclusions.The review search strategy is limited to public databases, Psychinfo and Medline. Other databases could have been addressed as well.

Reviewer #2 expresses concern that our use of only two public databases – PsychInfo and Medline – may limit our results. We deliberately selected these two databases to identify a comprehensive set of studies associated with the topic. Medline is the National Library of Medicine’s leading database, containing citations from 5,200 journals worldwide in biomedicine and health (https://www.nlm.nih.gov/medline/medline_overview.html).

PsychInfo is a top indexing database for psychological, social, and behavioral sciences and is used to supplement the findings from PubMed by capturing research that investigates the psychological factors that play a role in vaccine hesitancy. Studies are often cross-referenced between both databases.

We deliberately selected public (rather than proprietary) databases to ensure the reproducibility of our results. Moreover, this database selection allowed for a more refined search strategy to focus on empirical published peer-reviewed articles, excluding unpublished, grey literature, pre-publications, or dissertations.

We excluded Google Scholar from our database search as it does not offer a clear denominator as part of the search (i.e., it does not show us the total number of studies with text that contains key words), meaning it is unable to follow PRISMA guidelines. Google Search also uses a proprietary algorithm for producing results, which could change without any advance notification and thus undermine the reproducibility of our results.

We have added a sentence to our methods briefly describing why we chose these two databases.

A better-defined systematic review would have more clearly defined the search strategy. There is no mention of following PRISMA guidelines.

We have now clarified in the text (p. 9) that PRISMA guidelines were followed as part of our search and selection strategy. This is also clarified in the legend for Figure 2, which includes the PRISMA table describing the identification of relevant peer reviewed articles for this review.

The search strategy as presented lacks details, such as the procedure or individual records. Were titles and abstracts removed as a first step prior to final record selection? These gaps may ultimately have led to the few articles meeting the inclusion criteria for the first question and no articles addressing the second question.

Thank you for raising this concern. We have clarified the search strategy and review process in the text of the paper and in our figures. Specifically, we note that we carefully reviewed the full text of articles that contained the specified key words from our search strings, as opposed to only reviewing the titles and abstracts of articles. This maximized the likelihood that studies that met our criteria were not overlooked in our review.

In the Results section, a brief description of several studies' limitations was provided, but in the methods, it was indicated that cross-sectional correlational studies were to be targeted, but the lack of longitudinal studies was highlighted as a significant gap.

We have ensured that our methods has no language suggesting that our search was limited to cross-sectional correlational studies. Our search was designed to include all peer reviewed empirical studies that investigated the relationship between COVID-19 and HPV/HBV vaccine hesitancy, intention, or uptake. Our review found that few such studies existed and all were correlational in nature, which we discuss as a limitation of the research.

The Discussion section is short on implications and more of a 'Future Directions'.

We agree with the reviewer than in our previous draft, the implications of the summarized studies were insufficiently described. We have expanded our Discussion section to describe key takeaways from our systematic review for both RQ1 regarding spillover effects between COVID-19 vaccination and HPV/HBV vaccination and RQ2 regarding a possible relationship between COVID-19 misinformation on social media and HPV/HBV vaccination attitudes. We clearly indicate that the first paragraph addresses the implications of our findings for RQ1, while our second paragraph directly relates to RQ2 to facilitate readers’ comprehension of what we see as the main contributions of our review.

Finally, there is an extensive Future Research Directions on the role of social media that is based on no findings from the systematic and more of a polemic on its role in facilitating the dissemination of misinformation on vaccines. While this is interesting, the model presented in Figure 2 on the links among misinformation propagators and refuters does not really add to the overall impact of the paper, which should be based on the findings.

We appreciate this constructive critique. We now more clearly delineate between conclusions that are based soundly on the findings versus ideas that are based on the current work and expertise of authors. Specifically, our discussion first describes future research based on the systematic review to allow us to define necessary future research based on what our systematic review revealed about the state of the literature with regards to vaccine hesitancy and misinformation. We then elaborate on additional research directions to offer more innovative research ideas based on our own expertise in the interdisciplinary fields of computer science, epidemiology, and communication. This includes a shortened and greatly revised description of how we think Figure 1 represents an innovative method for studying the question of the implications of social media misinformation for vaccine attitudes and behaviors. We think these revisions greatly clarify the contribution of our work and thank the reviewer for this suggestion.

Recommendations for the authors:1. Introduction could be more concise and pointed, and many of the statements made are not directly relevant to the research questions.

We have streamlined the introduction to more narrowly focus on our research questions.

2. Expand the search strategy using consultation from a librarian and under PRIMA guidelines and registration. This will strengthen the fundamental search strategies and might aid in modifying the two objectives of the study.

We appreciate the encouragement to consider the best search strategy approach for our systematic review. We have described the rationale for our decision to limit our searches to the PubMed and PychInfo databases in our response to Reviewer #2, comment 1.

In response to Reviewer #2’s comments, we now more clearly identify our use of PRISMA guidelines in our search strategy. In accordance with best practices, we performed a bridge search through May 31, 2023 to incorporate any materials published between our previous search (on October 22, 2022) and our revisions. Using the same review process as described in the paper, we identified an additional 3 studies for RQ1 that met our criteria of empirically investigating the link between COVID-19 and HPV/HBV vaccine hesitancy, intention, or uptake, and 0 studies for RQ2 that met our criteria of empirically testing the relationship between exposure to COVID-19 vaccine misinformation on social media and HPV/HBV vaccine hesitancy, intention, or uptake.

We have revised the PRISMA figure to reflect the results across both research questions and describe the results of the bridge search in our methods section. Our revised Results section reports on the results of all of these studies identified in both the original and bridge searches.

3. The Methods should be more clearly defined in a stepwise fashion, e.g., including first a title and abstract screening to identify articles for full-text review.

Because our search yielded so few articles that potentially met criteria, there was no need to have a step of only reviewing titles and abstracts. We were able to review to review the full text of all articles that potentially met criteria for research questions one and two. We have revised our methods section to be clearer about our search and review strategy.

4. Be clear in the limits of your investigation- only cross-sectional studies and/or longitudinal studies and support these inclusion criteria.

We have ensured that our methods has no language suggesting that our search was limited to cross-sectional correlational studies. Our search was designed to include all peer reviewed empirical studies that investigated the relationship between COVID-19 and HPV/HBV vaccine hesitancy, intention, or uptake. Our review found that few such studies existed and all were correlational in nature, which we discuss as a limitation of the research.

5. In the discussion, please ensure that the implications of summarized studies are made clear, especially regarding any studies ultimately discovered regarding the role of social media information if found in a more expanded systematic review.

Please see our comments above with regards to the changes to the Discussion section to clarify the implications of our summarized studies.

6. Finally, ensure that the future directions are based soundly on the findings and any additional innovative ideas that spring from the author's own insights or experience.

Please see our comments above about the revisions to the future direction section, with special attention paid to distinguishing between those results based on the findings versus those that are based on our insights and experiences.